# Effects of Nanoclay on Mechanical and Dynamic Mechanical Properties of Bamboo/Kenaf Reinforced Epoxy Hybrid Composites

**DOI:** 10.3390/polym13030395

**Published:** 2021-01-27

**Authors:** Siew Sand Chee, Mohammad Jawaid, Othman Y. Alothman, Hassan Fouad

**Affiliations:** 1Laboratory of Biocomposite Technology, Institute of Tropical Forestry and Forest Products (INTROP), Universiti Putra Malaysia (UPM), Serdang 43400, Malaysia; joeychee1025@gmail.com; 2Department of Chemical Engineering, College of Engineering, King Saud University, P.O. Box 22452, Riyadh 11451, Saudi Arabia; 3Applied Medical Science Department, Community College, King Saud University, P.O. Box 10219, Riyadh 11433, Saudi Arabia; menhfef@ksu.edu.sa; 4Biomedical Engineering Department, Faculty of Engineering, Helwan University, Cairo 1790, Egypt

**Keywords:** hybrid composites, nanoclay, bamboo fibers, kenaf fibers, mechanical properties, dynamic mechanical properties

## Abstract

Current work aims to study the mechanical and dynamical mechanical properties of non-woven bamboo (B)/woven kenaf (K)/epoxy (E) hybrid composites filled with nanoclay. The nanoclay-filled BK/E hybrid composites were prepared by dispersing 1 wt.% nanoclay (organically-modified montmorillonite (MMT; OMMT), montmorillonite (MMT), and halloysite nanotube (HNT)) with high shear speed homogenizer followed by hand lay-up fabrication technique. The effect of adding nanoclay on the tensile, flexural, and impact properties of the hybrid nanocomposites were studied. Fractography of tensile-fractured sample of hybrid composites was studied by field emission scanning electron microscope. The dynamic mechanical analyzer was used to study the viscoelastic properties of the hybrid nanocomposites. BK/E-OMMT exhibit enhanced mechanical properties compared to the other hybrid nanocomposites, with tensile, flexural, and impact strength values of 55.82 MPa, 105 MPa, and 65.68 J/m, respectively. Statistical analysis and grouping information were performed by one-way ANOVA (analysis of variance) and Tukey method, and it corroborates that the mechanical properties of the nanoclay-filled hybrid nanocomposites are statistically significant. The storage modulus of the hybrid nanocomposites was improved by 98.4%, 41.5%, and 21.7% with the addition of OMMT, MMT, and HNT, respectively. Morphology of the tensile fracture BK/E-OMMT composites shows that lesser voids, microcracks and fibers pull out due to strong fiber–matrix adhesion compared to other hybrid composites. Hence, the OMMT-filled BK/E hybrid nanocomposites can be utilized for load-bearing structure applications, such as floor panels and seatbacks, whereby lightweight and high strength are the main requirements.

## 1. Introduction

The exploitation of our Earth and the deprivation of our environment has increased at a worrying rate for the past few decades. Due to this, there has been a lot of attention within society and the research community on the production of material with renewable resources. In the composites material industry, replacing the synthetic reinforcing fibers with natural fibers is one of the efforts and approaches used to reduce the impact on our Mother Earth. Given this, the natural fibers composites (NFCs) market worldwide has projected to grow by a compounded growth of 10.6% for the forecasted years from 2019 to 2025 [1]. The advantage of using lignocellulosic fibers, such as kenaf, bamboo, jute, hemp, and flax, in composites manufacturing is owing to their low density, low cost, relatively high strength, biodegradability, and their environmentally friendly quality. Recent technology advances in natural fibers composites have shown that synthetic fiber can be replaced by natural fibers in automotive, building, and packaging material applications [2,3,4,5].

Hybrid composites are produced from two or more types of reinforcing/filler incorporated in a polymer matrix [6]. Each reinforcer/filler has a feature that can surpass another reinforcer/filler limitations [7]. Hybridizing two types of natural fibers in a composite system is less frequent, but these are potentially beneficial materials given in relation to ecological issues. Kenaf and bamboo fibers have been reported to hybridize with different lignocellulosic fibers, such as coir, oil palm empty fruit bunch (OPEFB), jute, pineapple leaf, hemp, etc. Most of these findings show that hybrid composites exhibit improved mechanical properties [8,9,10], thermal properties [11], and moisture absorption behavior [12] which cannot be attained by single fiber-reinforced composites. Hanan et al. [9] studied the hybridization effects of OPEFB/kenaf fiber mats reinforced with epoxy composites, and reported that kenaf fiber-based composites renders better tensile and flexural properties, while OPEFB composites show better impact properties. Rihayat et al. [13] carried out a study with bamboo, pineapple leaf, and coir fibers in single and hybrid composites. They reported that bamboo/polyester composites exhibit the highest tensile strength (255 MPa), compared to pineapple leaf/polyester (116 MPa) composites and coir fiber/polyester (92 MPa) composites. Hence, hybrid composites with the highest weight fraction of bamboo fibers exhibit the highest tensile strength of 136 MPa.

Polymer nanocomposites (PNCs) are new materials developed from the technology breakthrough on nanotechnology and nanomaterial. Nano-sized fillers (nanoclay, graphene, carbon nanotube, etc.) are used to modify the polymer matrix to improve its intrinsic properties, like enhancement of mechanical properties, better barrier properties, good solvent, and fire resistance. Epoxy-based layered silicate nanocomposites, which are filled with montmorillonite (MMT), organoclay, saponite, and halloysite nanotubes (HNTs), have been widely reported, due to the ease of processing as well as its versatile applications in various fields. Besides, nanoclays are inexpensive compared to other nanomaterials and exhibit wide beneficial properties.

Hybridizing natural fibers together with nanoclay show great promising properties that can surpass the constraint of traditional composites, such as the hydrophilic nature of natural fibers, low heat, and fire-resistance. Recently, several research findings based on nanoclay/natural fibers hybrid polymer composites show improvement in mechanical properties. Saba et al. [14] carried out mechanical studies on nanofiller (oil palm nanofiller, MMT, and organically-modified MMT (OMMT))/kenaf/epoxy hybrid composites. Mechanical properties of the hybrid nanocomposites showed significant improvement in terms of tensile, impact strength, and elongation break. Ramesh et al. [15] studied the effect of different MMT loadings on treated kenaf fiber/polylactic acid hybrid biocomposites. They concluded that 1% MMT clay loading into PLA/treated kenaf fiber renders improved mechanical properties. Adamu et al. [16] reported on the improvement of the modulus of elasticity and modulus of rupture of bamboo/polyvinyl alcohol/clay nanocomposites. Kushwaha et al. [17] also found improvement in the tensile and flexural properties of epoxy/bamboo mat/MMT clay hybrid composites. The elastic modulus of the hybrid nanocomposites increases by 16.25% at 1% clay loading.

The dynamic mechanical analysis offers useful data on the viscoelastic and damping behavior of polymer composite materials. Besides, these dynamic parameters produce helpful insight about the interfacial bonding between the different constituent in a polymer composite [18], crosslinking density [19], and phase changes [20]. Several researchers corroborated on the improvement of dynamic mechanical properties of hybrid composites reinforced solely with natural fibers. Asim et al. [21] carried out a study on silane-treated kenaf/pineapple leaf fibers phenolic hybrid composites. The silane-treated hybrid composites show enhancement in storage modulus as compared to the untreated hybrid composites. Sathiskumar [22] reported on the dynamic properties of snake grass fiber-reinforced polyester composites. A balance of properties of storage modulus and damping behavior was achieved for a 30% weight fraction of fiber-containing composites having 30 mm fiber length. The addition of nanofiller often changed the relaxation behavior of macromolecular polymer chains [23]. Rajesh et al. [24] reported that the incorporation of nanoclay improved the storage modulus and loss modulus but lowered the loss factor (tan delta) of banana/jute/nanoclay polyester composites, due to the improved engagement between polyester, nanoclay and banana/jute woven fibers.

From the literature reviewed, it can be summarized that nanoclay/natural fibers hybrid nanocomposites have attained great interest owing to their improved properties such as mechanical properties, barrier properties, thermal, and fire performance. However, to date, there is still limited work reported on the effect of nanoclay-modified epoxy on hybrid composites reinforced with two types of mat form natural fibers. Thus, in this study, the bamboo and kenaf fiber mat was used to prepare hybrid composites reinforced in a nanoclay-modified epoxy matrix. The main objective of the current study is to study the effects of the addition of nanoclay filler (organically-modified MMT (OMMT), montmorillonite (MMT), halloysite nanotube (HNT)) on the mechanical (tensile, flexural, and impact properties) and dynamic mechanical properties of the hybrid nanocomposites and compare the same to unfilled hybrid composites. It is anticipated that the incorporation of nanoclay would improve the mechanical and dynamic mechanical properties without affecting the mass of the composites. Hence, the outcome of this study could be utilized as load-bearing structure applications in the automotive, building, and construction sectors.

## 2. Materials and Method

### 2.1. Materials and Chemicals

The woven kenaf fiber mat (Figure 1a) was supplied by ZKK Sdn. Bhd, Selangor, Malaysia. The density and unit area density is 1.44 g/cm^3^ and 0.06 g/cm^2^. Shijiangzhuang Bi Yang Technology Co. Ltd., Hebei, China supplied the non-woven bamboo mat (Figure 1b). The density and unit area density of the bamboo mat is 1.20 g/cm^3^ and 0.08 g/cm^2^. The cellulose, hemicellulose, and lignin content of the fibers were determined by neutral detergent fiber (NDF), acid detergent fiber (ADF), and lignin analysis, respectively. The cellulose content of kenaf and bamboo fibers is 65.7% and 72.6%. Hemicellulose content is 17.8% and 11.1%; lignin content is 6.0% and 9.5%, respectively. The fiber mats were pre-cut into the size of 240 mm × 120 mm × 3 mm according to the mold size used for the fabrication. The nanoclays used in this work were all purchased from Sigma Aldrich, Selangor, Malaysia. The MMT possesses a mean particle size of ≤25 µm and a bulk density of 600 to 1100 kg/m^3^. The OMMT (Nanomer I.31PS, Nanocor, Illinois, United States) was chemically modified with 15% to 35% of octadecyl ammonium (ODA) and 0.5% to 5% of 3-aminopropyltriethoxy silane (APTES). It possesses a mean particle size of 14 to 18 µm and a bulk density of 250 to 300 kg/m^3^. The HNT possesses a tube-like structure with a diameter of 30 to 70 mm and a length of 1 to 3 µm. The natural fibers and nanoclays that were used in this work were used as received without any further purification.

### 2.2. Fabrication of Hybrid Composites

The amount of nanoclay used in this work was fixed at 1 wt.% and the preparation of epoxy–nanoclay mixture was according to our previous work [25]. A mechanical stirrer was used to pre-mix the DGEBA (diglycidyl ether of bisphenol-A) epoxy resin and nanoclay with the slow addition of nanoclay into the epoxy resin. To enhance better dispersion of the nanoclay, the epoxy–nanoclay mixture was subjected to a high shear speed of 10,000 rpm by using a T-25 Ultra Turrax homogenizer (IKA, Staufen, Germany). It was mixed for 30 min under an ice water bath to reduce the excessive heat generation caused by the high shear speed. To reduce the air trapped bubbles, the mixture was then degassed for 30 min before the addition of the cycloaliphatic amine hardener. The fibers-reinforced epoxy–nanoclay hybrid nanocomposites were fabricated by hand lay-up technique by stacking the bamboo and kenaf mat layer-by-layer in a stainless steel mold. The total fiber loading was controlled at 40 wt.% with 1:1 ratio of kenaf fiber to bamboo fiber, as per our previous work [26]. The nanoclay–epoxy–hardener mixture was poured slowly into the mold to ensure the fibers were fully impregnated within the mixture. Any air bubbles were removed slowly with a roller before closing the mold. The mold was left at room temperature for 24 h for curing then was post-cured at 105 °C for 5 h. A controlled sample that did not contain nanoclay was fabricated for comparison purposes.

### 2.3. Characterization

The tensile properties of the hybrid composites were determined as per ASTM D3039 by using a universal testing machine, model 5566 (INSTRON, Massachusetts, United States). Five specimens of each group of composites were cut to 120 mm × 20 mm × 3 mm in size. The gauge length of the hybrid composites was 60 mm and tested at a constant cross-head speed of 2 mm/min with a 10 kN load cell. The flexural properties of the hybrid composites were tested using a three-point bending test according to Procedure A in ASTM D790 by using 5 kN Universal Testing Machine model AI-3000 Gotech (UTM, Gotech, Taichung City, Taiwan). Five specimens of each group of composites were prepared to the dimensions 127 mm × 12.7 mm × 3 mm. The support span length of each group of specimens was determined separately, as outlined in ASTM D790 standards, where the support span, depth ratio of the specimens shall maintain at 16:1. The testing speed was determined according to Equation (1).
(1)$$R=ZL^{2}/6d$$
where *Z* = strain rate of 0.01 mL/min, *L* = support span, mm, *d* = depth of beam, mm, and *R* = testing speed, mm/min, as per defined in Procedure A, ASTM D790.

The Izod pendulum impact resistance of the hybrid composites was tested by using the Gotech GT-7045 (Taichung City, Taiwan) impact tester as per ASTM D256. The specimens were prepared in the dimensions of 63 mm × 13 mm × actual specimen thickness (3 to 3.5 mm) with a notch angle of 45° and the depth of 2.5 mm. The pendulum speed used in this study was fixed at 3.5 m/s. All the mechanical testing specimens were placed in a preconditioning chamber at the temperature of 23 ± 3 °C and relative humidity 50 ± 10% for 40 h before analysis. Each group of composites containing 5 specimens were tested and the average value was reported. “Minitab 19” (Minitab, State College, PA, USA) software was used to carry out the statistical analysis. One-way ANOVA (analysis of variance) analysis was used to evaluate the significant difference between the average mechanical properties of the different hybrid composites. The Tukey method was used to analyze the grouping information of the hybrid composites. The fractography analysis was carried out on tensile-fractured specimens by using the FEI FESEM (Field Emission Scanning Electron Microscope) (Nova NanoSem, Hillsboro, OR, USA). A Mettler Toledo dynamic mechanical analyzer (DMA 1, Greifensee, Switzerland) was used to study the viscoelastic behavior of the specimens. The measurements were carried out following ASTM D5023-01 by using the single cantilever mode. The samples were studied under a temperature ramp method from 25 to 150 °C. The ramp rate, oscillating frequency, and deformation amplitude were fixed at 5 °C/min, 1 Hz, and 10 µm, respectively.

## 3. Results and Discussion

### 3.1. Tensile Properties

Figure 2 presents the stress vs strain curve of the hybrid composites. All the curves behave linearly up to the failure of the composites. Controlled samples (BK/E hybrid composites) demonstrate the lowest stress; however, it could withstand the highest strain among the fabricated hybrid composites. On the other hand, the nanoclay-filled hybrid composites revealed a higher stress value but lower strain value compared to the controlled samples. This indicates the nanoclay-filled hybrid composites exhibit higher strength but behave more brittle in nature compared to the controlled samples. BK/E-OMMT yields the highest stress, followed by BK/E-MMT and BK/E-HNT. Figure 3 illustrates the tensile strength and Young’s modulus of the fabricated hybrid composites. The inclusion of nanoclay improved the tensile strength and Young’s modulus. The tensile strength of the BK/E is 38.6 MPa. With the addition of MMT, HNT, and OMMT, the tensile strength was improved by 19.12%, 13.58%, and 44.5%, respectively. On the other hand, Young’s modulus of the BK/E-MMT (2.39 GPa), BK/E-HNT (2.35 GPa), and BK/E-OMMT (2.51 GPa) was improved by 87.0 7%, 45.81%, and 96.54%, respectively, in relative to BK/E (1.27 GPa). BK/E-OMMT shows remarkable improvement in tensile properties among all hybrid nanocomposites due to the presence of organic modifiers, which enhanced better interlayer spacing, and well-dispersed nanoclay, creating strong interfacial adhesion strength between the different constituents in a hybrid nanocomposite. From our previous work [25], the interfacial adhesion strength between the epoxy and OMMT can be improved with the chemical reaction between the amine group (–NH2) of 3-aminopropyltriethoxy silane and the oxirane ring of the DGEBA monomer. This led to efficient stress or load transfer from matrix to fibers, resulting in a delay in the fracture or crack initiation mechanism. Besides, non-chemically-treated MMT and HNT are difficult to disperse in epoxy resin, due to their naturally hydrophilic nature, which makes it incompatible with non-polar epoxy resin. This will lead to lower mechanical properties performance. Similar observations were also reported by [14,27]. The microstructure studies on the interaction and distribution of the nanoclays and epoxy resin are supported with the outcome of the results from our previous work [25,28]. The structural and morphology study, by using WAXS (wide angle x-ray scattering) and FESEM from our previous works [25], reveals that OMMT nanoclay dispersed more evenly in the epoxy matrix, while high agglomeration is observed in the MMT–epoxy and HNT–epoxy mixtures.

One-way ANOVA statistical analysis was used to evaluate the significant difference between the average tensile strength and Young’s modulus of the hybrid composites (Table 1). The difference of tensile strength and modulus between the groups (BG) and within the groups (WG) is determined. The difference between groups refers to the significance of the treatment effects, in our case it refers to the effect of inclusion of different kinds of nanoclay, while difference within groups refers to the errors occurring. The ratio between differences of BG to WG leads to F-value determination, which is an important indicator of the treatment effects. An F-value greater than 1 indicates positive treatment effects; a greater value shows greater treatment effects. The F-value obtained for Young’s modulus (153.1) is greater than tensile strength (21.30), indicating that the inclusion of nanoclay improved Young’s modulus more than tensile strength. This may be ascribed to the high moduli properties of clay particles when it is dispersed in the nanoscale level in the polymer matrix [29]. *p*-value is used to indicate the differences between the means are statistically significant. We reject the null hypothesis with proof of the *p*-value resulting for tensile strength and modulus being less than 0.05. This indicates the differences between mean tensile strength and modulus of the different types of hybrid composites are statistically significant at a 95% confidence level. The grouping information was then analyzed according to the Tukey method at the 95% confidence level and the results are tabulated in Table 2. The groups that have the same letter indicate that there is no evidence of a difference for that pair. The Tukey analysis reviewed that the tensile strength between BK/E-MMT–BK/E-HNT and BK/E-HNT–BK/E are not significantly different. The tensile strength of BK/E-OMMT is significantly different from all other composites. Besides, Young’s modulus of BK/E-OMMT, BK/E-MMT, and BK/E-HNT are significantly different compared to BK/E-Epoxy.

### 3.2. Flexural Properties

The maximum bending stress used to break a beam under a three-point bending load refers to the flexural strength of a material [30]. The flexural strength of the hybrid composites is shown in Figure 4. The incorporation of nanoclay enhanced the flexural strength of BK/E. BK/E-OMMT recorded the highest mean flexural strength value at 105 MPa, followed by BK/E-MMT and BK/E-HNT with mean flexural strength values of 78 and 75 MPa, respectively. BK/E recorded the lowest mean tensile strength among all hybrid composites with a value of 65 MPa. Flexural modulus refers to the ratio of stress and strain in a materials’ linear elastic region. It provides insight into the material stiffness or resistance to bending when stress under a three-point bending flexural test technique. The flexural modulus of the different types of hybrid composites is presented in Figure 5. Maximum improvement was observed on BK/E-OMMT by an increment of 102%, as compared to BK/E, while BK/E-MMT and BK/E-HNT increase by 61.6% and 60.9%, respectively.

Nanoclays possess a high aspect ratio with a relatively large surface area, as compared to its thickness in the order of nanometer. MMT has an aspect ratio ranging from 100 to 1000 [31], while HNT possesses an aspect ratio from 10 to 50 [32]. The improvement of flexural strength and modulus can be explained by the large aspect ratio of nanoclay, which is able to fill up the gaps between the reinforcing fibers matrix [27], thus reducing the void content which can behave as the weak spot of the composites. The presence of organically-modified MMT is known to improve the interfacial adhesion strength between epoxy matrix, bamboo, and kenaf fibers leading to improved interfacial bonding of the hybrid nanocomposites. Strong interfacial adhesion strength renders stronger and more even stress distribution. Additionally, the intrinsic property of clay (high moduli) provides an excellent compression property, leading ultimately to the increase in flexural properties. Rafiq et al. [33,34] also reported that strong interfacial bonding between reinforcement (glass fibers and organically-modified nanoclay) and epoxy matrix exhibit higher flexural properties. However, unmodified MMT and HNT are intrinsically hydrophilic and unable to disperse well in a non-polar polymer matrix. Thus, the relatively lower flexural properties observed on BK/E-MMT and BK/E-HNT are due to the aggregation of the nanoclays and weak interfacial bonding.

Table 3 and Table 4 tabulated the findings of the statistical analysis by one-way ANOVA and grouping information by Tukey method. As expected, the F-value obtained in flexural modulus (153.1) is higher compared to flexural strength (63.21), indicating nanoclay created better improvement in the flexural modulus due to possessing a greater elastic modulus compared to the epoxy matrix [35]. The *p*-value of the flexural strength and modulus is lower than 0.05, which rejects the null hypothesis. ANOVA analysis indicates that significant differences exist in all the hybrid composites in regards to their flexural strength and modulus at a 95% confidence level. On the other hand, the Tukey analysis reveals that the flexural strength and modulus of all three types of nanoclay-filled hybrid nanocomposites are significantly different from the controlled sample. However, BK/E-MMT and BK/E-HNT show the same grouping on their flexural strength and modulus, indicating no significant difference exists between them.

### 3.3. Impact Strength

The ability of a substance to absorb an abrupt impact load is referred to as the impact strength in the unit of J/m [36]. The impact performance of a composite material relies on several aspects such as the nature of the constituent, the interfacial bonding of fillers/fibers/matrix, and the strength and structure of the reinforcing fibers [36,37]. The effects of adding nanoclay on the impact strength of the hybrid composites are illustrated in Figure 6. The recorded impact strength for BK/E is 42.46 J/m and its impact behavior is mainly dominated by the woven kenaf structure compared to non-woven bamboo mat. Alavudeen et al. [37] and Safwan et al. [38] reported that woven structure fibers render better impact strength compared to random fibers orientation. The addition of nanoclay increased the impact strength of the hybrid composites due to the improved interfacial adhesion strength between the matrix and reinforcing agents. The enhancement of 21.8%, 19.6%, and 54.7% impact strength of BK/E-MMT, BK/E-HNT, and BK/E-OMMT, as compared to BK/E, was observed. It was also seen that the impact strength of BK/E-OMMT increased by 27.0% and 24.4% as compared to BK/E-MMT and BK/E-HNT, respectively. This can be attributed to the fact that the incorporation of OMMT induces better energy absorption due to the strong interfacial bonding, which hinders the initial break, break pinning mechanism and its extension within the composite under the applied force [14]. The results are in line with nanoclay-filled polylactic acid/polycaprolactone/oil palm mesocarp fibers hybrid composites [39] and OMMT/kenaf-reinforced epoxy hybrid composites [14].

One-way ANOVA statistical analysis was carried to evaluate the significant difference between the average impact strength of the different hybrid composites (Table 5). The *p*-value is smaller than 0.05, indicating that a substantial difference exists between the average impact strength of the hybrid composites at a 95% confidence level. Tukey analysis (Table 6) revealed that BK/E-OMMT falls under group A, while BK/E-MMT, BK/E-HNT, and BK/E fall under group B, indicating BK/E-OMMT is significantly different compared to the other hybrid composites. However, no significant differences were found between BK/E-MMT, BK/E-HNT, and BK/E.

### 3.4. Fractography Study on Tensile-Fractured Specimens

The surface of tensile-fractured specimens was investigated by FESEM; the micrographs are displayed in Figure 7. The micrographs show evidence of delamination and microcracks around fibers and the matrix, and traces of fibers pull out and voids were observed. These observations may be related to the poor interfacial bonding between the fibers and matrix which leads to poor stress transfer from matrix to fiber, thus resulting in lower mechanical properties. Delamination is observed on BK/E (Figure 7a) between bamboo and kenaf fibers layers, ultimately leading to the advanced matrix crack. The woven kenaf mat consists of fibers weaving in warp and weft directions. When tensile stress is applied in the loading direction, stretching occurs in the longitudinal fibers, at the same time the transverse direction fibers are also under stress and tend to straighten. This will induce stress concentration at the interface, leading to microcracks in the polymer, which then propagate in a transverse direction causing, ultimately, fibers and matrix fracture. Similar observations are also found on BK/E-MMT (Figure 7b) and BK/E-HNT (Figure 7c), whereby microcracks are noticed around kenaf fibers; voids due to fiber pull out and trapped air bubble are observed too. The addition of nanoclay showed improved interfacial interactions, with moderate delamination and microcracks, as compared to BK/E. On the other hand, the FESEM image of BK/E-OMMT (Figure 7d) shows comparatively lesser voids, microcracks and fibers pull out due to strong fiber–matrix adhesion. These observations are in line with the author’s earlier published work on the study of the void contents of the hybrid composites [40]. BK/E-OMMT exhibits the lowest void content (1.28%), followed by BK/E-MMT (3.77%), BK/E-HNT (4.66%), and BK/E (5.50%). The fibers undergo more breakage instead of the fiber pulled out from the matrix surface during the load applied, indicating better stress transfer from matrix to fibers. Besides, the lower mechanical performance observed on BK/E-MMT and BK/E-HNT is also related to the agglomeration of the nanoclay, as shown in Figure 7e,f.

### 3.5. Dynamic Mechanical Analysis (DMA)

#### 3.5.1. Storage Modulus (E′)

Storage modulus characterizes the elasticity behavior of a material when it is subjected to sinusoidal stress. It provides information on the dynamic mechanical properties of a material such as stiffness [25], load-bearing capacity [41], cross-link density [42], and interfacial strength between fiber and matrix [43]. The effect of incorporating a different type of nanoclay on the storage modulus (E′) of the hybrid composites is displayed in Figure 8 and the data are tabulated in Table 7. The storage modulus of the hybrid composites before and after the glass transition region was improved with the addition of nanoclay. The E′ of the hybrid composites in the glassy region was improved by 98.4%, 41.5%, and 21.7% with the addition of OMMT, MMT, and HNT nanoclay, respectively. This may be ascribed to the toughening effect by the nanoclay, which limited the movement of the polymer chain [44]. Besides, the stress can be effectively transferred from the nanoclay-modified epoxy matrix to the reinforcement fibers owing to the improved interfacial adhesion strength between fibers and matrix [45,46]. BK/E-OMMT displayed the highest E′ among all hybrid composites. This can be ascribed by the fact that organically-modified nanoclay stimulated the formation of intercalated/exfoliated nanocomposites. The mobility of the polymer chain is being constrained as the epoxy polymer gets intercalated in between the layered OMMT. Similar findings were reported by Saba et al. [45] that OMMT/kenaf/epoxy hybrid composites showed prominent enhancement in E′ compared to OPEFB (oil palm empty fruit bunch)/kenaf/epoxy hybrid composites and MMT/kenaf/epoxy hybrid composites. The relatively lower E′ value observed in BK/E-MMT and BK/E-HNT is due to the immiscible mixing of epoxy resin with unmodified nanoclay, which leads to heterogeneity in the resultant samples. As a result of this, big tactoid formations occurred due to agglomeration of the unexfoliated clay [47]. Again, we observed that the E′ in the rubbery region was improved, with the highest E′ value recorded by BK/E-OMMT (173 MPa), followed by BK/E-MMT (148 MPa), BK/E-HNT (139 MPa), and BK/E (133 MPa). In the rubber region, the E′ value of all hybrid nanocomposites was also improved, indicating a strong fiber-nanoclay matrix interfacial bonding, which allowed efficient stress distribution from the matrix to fibers [48]. The storage modulus curve also revealed the glass transition (*T_g_*) temperature of the composites where a sudden change of modulus was observed around 60–80 °C. *T_g_* is a transition over a range of temperatures from a glassy state to a rubbery state in an amorphous material. In the glassy region, the mobility of the molecules in the epoxy matrix is constrained due to the tightly packed molecule arrangement. Thus, the material behaves as stiff and rigid resulting in a high modulus value. Above the glass *T_g_* temperature, the free volume of the epoxy matrix increases as the tightly-packed molecule arrangement collapses, which allows higher molecular mobility. This occurs as the glass–rubber state transition with E′ decreases.

#### 3.5.2. Loss Modulus

Loss modulus (E″) indicates the viscous behavior of a material when subjected to an oscillating stress cycle [49]. A high loss modulus indicates a higher ability of dissipated energy and, hence, marks better damping properties to reduce damaging forces caused by mechanical energy. Figure 9 presents the influence of adding nanoclay on E″ of the hybrid composites as a function of temperature. All hybrid composites attained a maximum peak height in the *T_g_* region; the data are tabulated in Table 7. As shown in Figure 8, the glass transition region falls in the temperature range of 60 to 80 °C. In the glassy region, the material is stiff and rigid, thus its loss modulus is low and constant. However, when it passes through the glass transition region, changing from the glassy to the rubbery state, the viscous behavior of the material raises intensely. This can be explained by the molecular segmental motion in the polymer chain overlap with the mechanical deformation, resulting in high internal friction and non-elastic deformation [26]. Thus, results in high dissipation energy, with the loss modulus reaching maximum peak height; this is denoted as the glass transition (*T_g_*) temperature of the system. After reaching its maximum peak height, the molecules are now in a more relaxed phase, thus reducing the internal friction, resulting in the drop of the loss modulus. BK/E exhibit lowest E″ value (69 MPa), with the addition of nanoclay increases the E″ value of the BK/E-OMMT, BK/E-MMT, and BK/E-HNT by 159%, 65.2%, and 53.6%, respectively. This indicates that nanoclay addition has induced higher internal friction and lead to higher energy dissipation. Similar findings were also reported by [45,50]. The highest E″ value of BK/E-OMMT can be ascribed by the fact that polar and non-polar groups of the organically-modified MMT induced better interfacial interaction between the components leading to improve E″ value. Hazarika et al. [50] reported on the enhancement of the E′ and E″ with the incorporation of treated nanoclay in wood polymer composites. On the other hand, BK/E-MMT and BK/E-HNT displayed lower E″ value than BK/E-OMMT, but higher than BK/E due to the poor interfacial interaction between the components. Both E′ and E″ behavior of all the hybrid composites follow a similar trend (BK/E-OMMT > BK/E-MMT > BK/E-HNT > BK/E). Thus, we can also anticipate that the complex modulus (E*), which is the sum of E′ and E″, will improve with nanoclay addition and follow a similar trend. The higher E* also indicated higher possibilities of the higher tensile properties [45] in BK/E-OMMT, followed by BK/E-MMT, BK/E-HNT, and BK/E. This trend correlates well with earlier discussion on the tensile properties of the hybrid composites.

#### 3.5.3. Tan Delta

Tan delta curve refers to the ratio of loss and storage moduli (E′′/E′); it is known as the loss factor or damping parameter. A high tan delta value reveals that a material exhibits a high non-elastic strain component. Contrary, a low value suggests the material is more elastic. The tan delta curve indicates the energy dissipation as heat during each deformation cycle. Figure 10 illustrates the effect of adding nanoclay on the damping properties (tan delta) of the hybrid composites as a function of temperature. The tan delta peak height (Table 7) of nanoclay-filled hybrid composites were found to decrease compared to BK/E (0.21). This indicates B/K/E possesses higher damping properties with higher non-elastic deformation and high energy dissipation. The addition of nanoclay reduces the magnitude of the peak height. This can be attributed to the interlocking mechanism between the nanoclay, fibers and epoxy, which restricted the polymer chain movement [46]. BK/E-OMMT exhibits the lowest tan delta peak height among all hybrid composites, indicating strong interfacial interaction, resulting in lesser energy dissipation at the interface [50,51].

## 4. Conclusions

In this study it can be concluded that the bamboo fiber mat/kenaf mat fiber-reinforced nanoclay-modified epoxy hybrid nanocomposites were successfully prepared. The effect of adding OMMT, MMT, and HNT on the mechanical, dynamic mechanical properties and fractography analyses of the bamboo mat/kenaf and mat/epoxy composites was studied. The following conclusions were drawn from the analysis.

The tensile, flexural, and impact properties of the hybrid composites are improved with the inclusion of nanoclay. BK/E-OMMT hybrid composites exhibit the best mechanical performance among all hybrid nanocomposites, followed by BK/E-MMT and BK/E-HNT.BK/E-OMMT hybrid composites also show excel E′, E″, and tan delta values compared to other hybrid composites. Besides that, the relatively lower tan delta peak observed on BK/E-OMMT indicates strong interfacial bonding between fibers and the matrix.The FESEM images on the tensile-fractured samples confirmed that the addition of OMMT reduced void contents and showed strong fiber–matrix adhesion with reduced fiber pull out, delamination, and microcracks. However, the addition of MMT and HNT nanoclay exhibited agglomeration, higher void contents, delamination, and microcracks between the fiber–matrix interface.

Overall, it is concluded that the addition of OMMT nanoclay is an effective reinforcement, as it displays enhanced mechanical and dynamic mechanical properties with relation to MMT and HNT nanoclays.

## Figures and Tables

**Figure 1 polymers-13-00395-f001:**
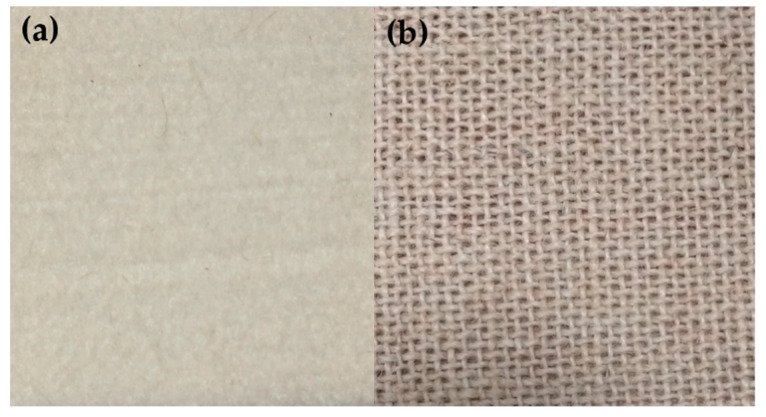
Photographic images of (**a**) non-woven bamboo mat; (**b**) woven kenaf mat.

**Figure 2 polymers-13-00395-f002:**
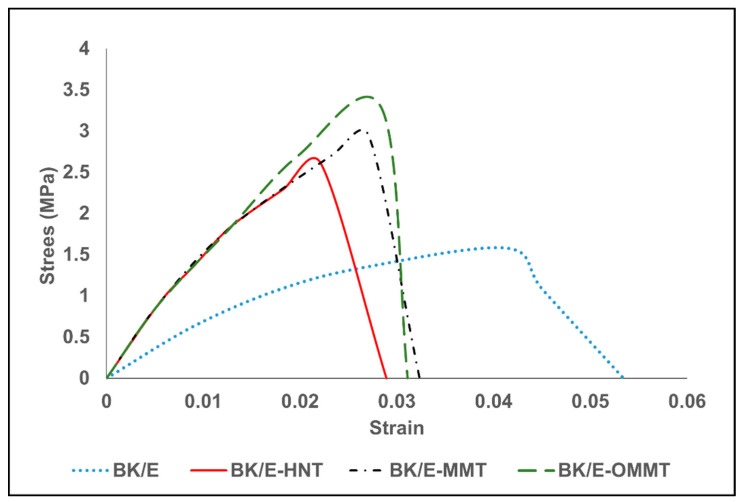
Tensile stress vs. strain curve of the of the hybrid composites.

**Figure 3 polymers-13-00395-f003:**
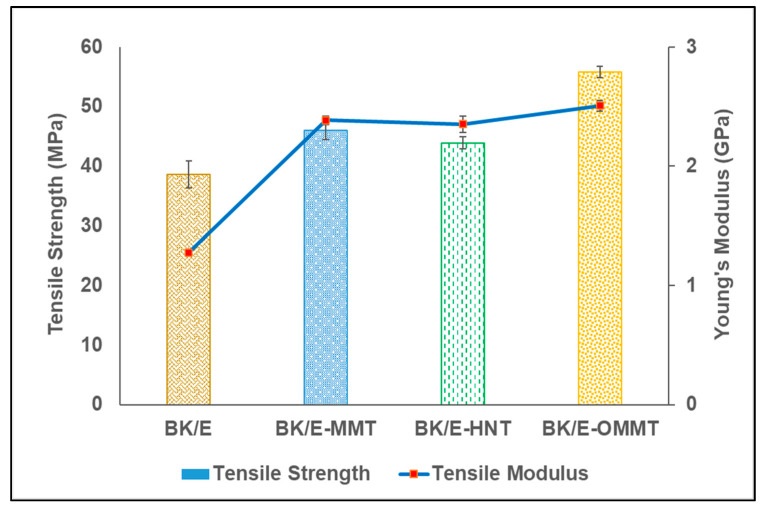
Tensile strength and Young’s modulus of the hybrid composites.

**Figure 4 polymers-13-00395-f004:**
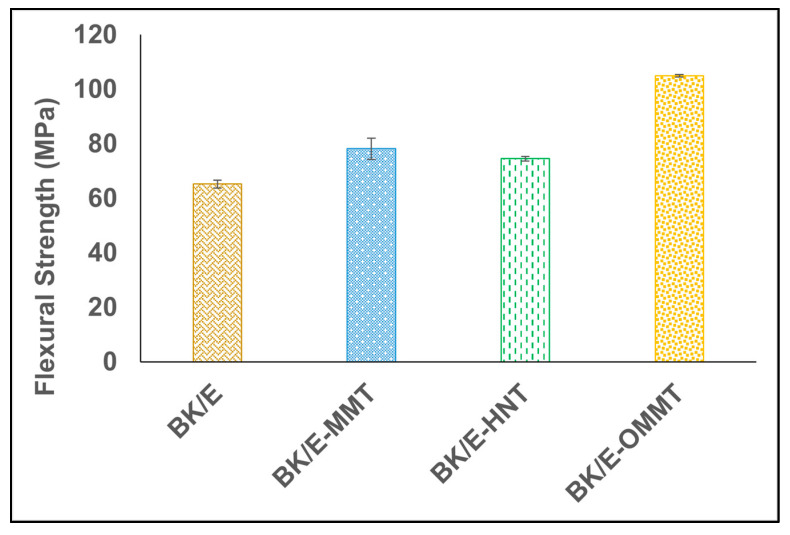
Flexural strength of the hybrid composites.

**Figure 5 polymers-13-00395-f005:**
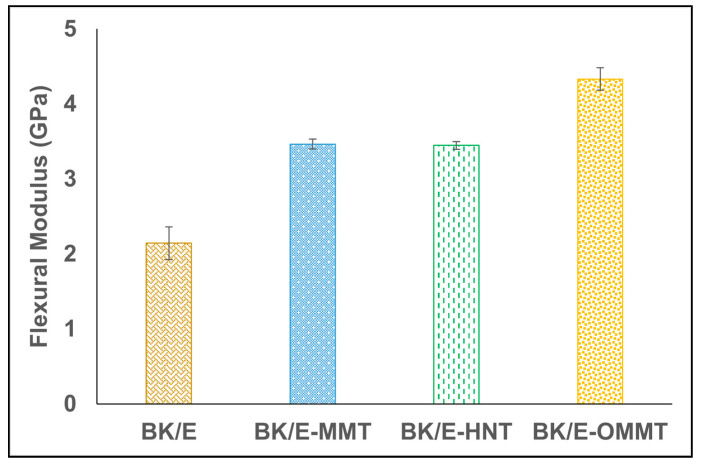
Flexural modulus of B/K/nanoclay-reinforced epoxy hybrid nanocomposites.

**Figure 6 polymers-13-00395-f006:**
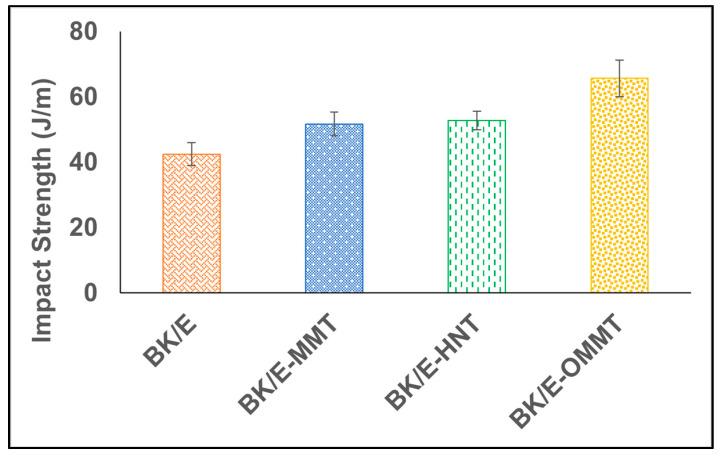
Impact strength of B/K/nanoclay-reinforced epoxy hybrid nanocomposites.

**Figure 7 polymers-13-00395-f007:**
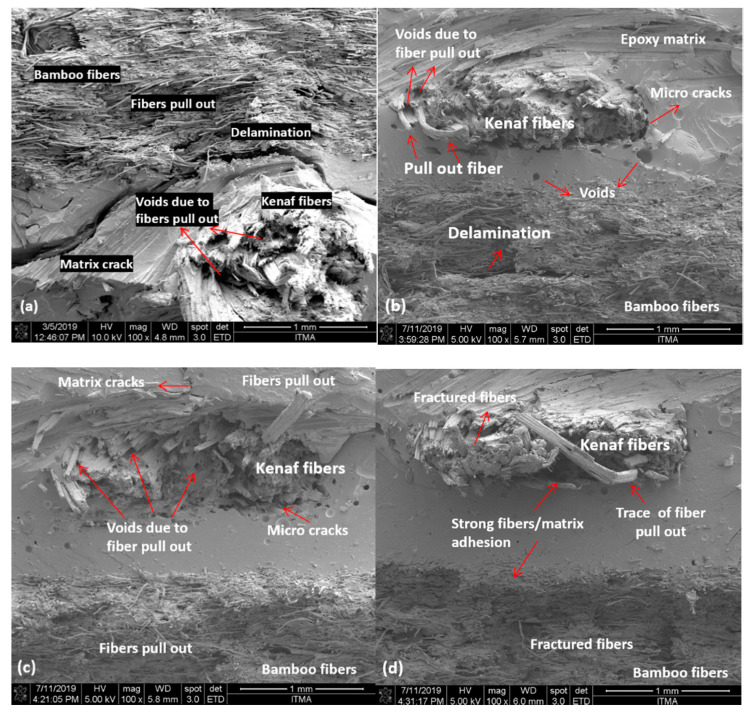
FESEM images of tensile-fractured samples. (**a**) BK/E; (**b**) BK/E-MMT; (**c**) BK/E-HNT; (**d**) BK/E-OMMT; (**e**) agglomeration observed in BK/E-MMT; (**f**) agglomeration observed in BK/E-HNT.

**Figure 8 polymers-13-00395-f008:**
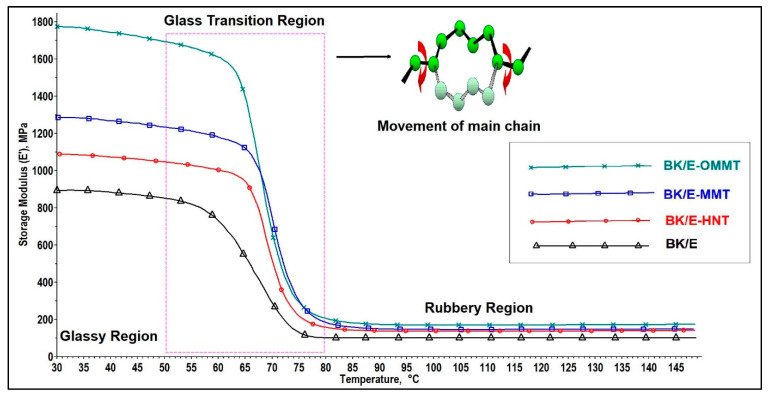
Effect of nanoclay on the storage modulus of the hybrid composites.

**Figure 9 polymers-13-00395-f009:**
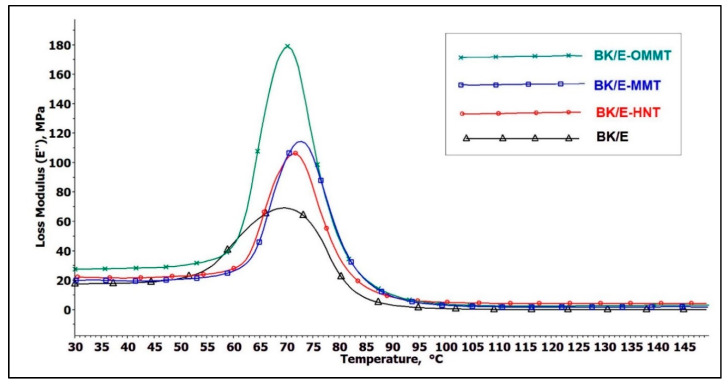
Effect of nanoclay on the loss modulus of the hybrid composites.

**Figure 10 polymers-13-00395-f010:**
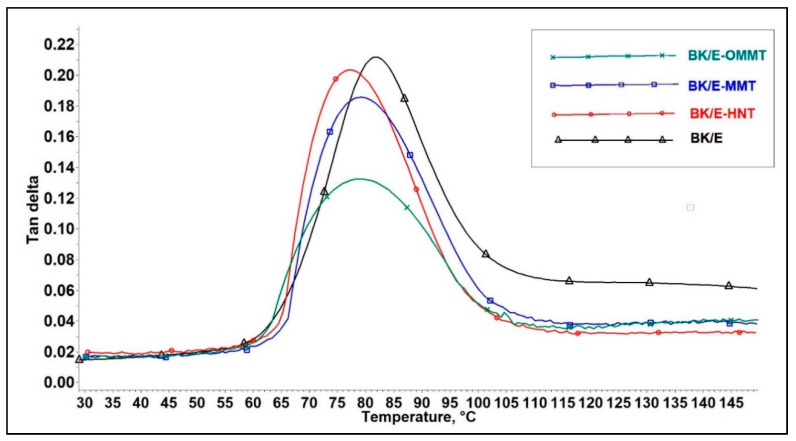
Effect of nanoclay on the tan delta of the hybrid composites.

**Table 1 polymers-13-00395-t001:** One-way ANOVA data analysis on tensile strength and Young’s modulus.

Properties	Source of Difference	*SS*	*df*	*MS*	F-Value	*p*-Value
Tensile Strength	Between Groups (BG)	776.7	3	258.9	21.30	0.000
	Within Groups (WG)	194.5	16	12.16		
Young’s Modulus	Between Groups (BG)	4.938	3	1.646	153.1	0.000
	Within Groups (WG)	0.172	16	0.011		

*SS*: Sum of square; *df*: Degree of freedom; *MS*: Mean square.

**Table 2 polymers-13-00395-t002:** Grouping information analysis by Tukey method on the mean tensile strength and Young’s modulus of the hybrid composites.

Hybrid Composites	Mean Tensile Strength, MPa	Grouping	Mean Young’s Modulus, GPa	Grouping
BK/E-OMMT	55.82	A	2.507	A
BK/E-MMT	46.01	B	2.386	A
BK/E-HNT	43.87	BC	2.353	A
BK/E	38.62	C	1.275	B

The groups that have the same letter (A, B, C) indicate that there is no evidence of a difference for that pair.

**Table 3 polymers-13-00395-t003:** One-way ANOVA data analysis on flexural strength and modulus.

Properties	Source of Difference	*SS*	*df*	*MS*	F-Value	*p*-Value
Flexural Strength	Between Groups (BG)	4361.1	3	1454	63.21	0.000
	Within Groups (WG)	368.0	16	23.00		
Flexural Modulus	Between Groups (BG)	12.22	3	4.074	153.1	0.000
	Within Groups (WG)	1.55	16	0.097		

*SS*: Sum of square; *df*: Degree of freedom; *MS*: Mean square.

**Table 4 polymers-13-00395-t004:** Grouping information analysis by Tukey method on the mean flexural strength and modulus of the hybrid composites.

Hybrid Composites	Mean Flexural Strength, MPa	Grouping	Mean Flexural Modulus, GPa	Grouping
BK/E-OMMT	104.98	A	2.507	A
BK/E-MMT	78.20	B	2.386	B
BK/E-HNT	74.58	B	2.353	B
BK/E	65.24	C	1.275	C

The groups that have the same letter (A, B, C) indicates that there is no evidence of a difference for that pair.

**Table 5 polymers-13-00395-t005:** One-way ANOVA data analysis on impact strength.

Properties	Source of Variation	*SS*	*df*	*MS*	F-Value	*p*-Value
Impact Strength	Between Groups (BG)	1201.9	3	400.6	10.19	0.001
	Within Groups (WG)	589.5	15	39.30		

*SS*: Sum of square; *df*: Degree of freedom; *MS*: Mean square.

**Table 6 polymers-13-00395-t006:** Grouping information analysis by Tukey method on the mean impact strength of the hybrid composites.

Hybrid Composites	Mean Impact Strength (J/m)	Grouping
BK/E-OMMT	65.67	A
BK/E-MMT	52.78	B
BK/E-HNT	51.70	B
BK/E	42.46	B

The groups that have the same letter (A, B, C) indicate that there is no evidence of a difference for that pair.

**Table 7 polymers-13-00395-t007:** Results summary of DMA analysis of the hybrid composites.

Composites	E′ at 25 °C (MPa)	E′ at 120 °C (MPa)	Peak of E″ (MPa)	Peak Height of Tan Delta
BK/E-OMMT	1776	173	179	0.13
BK/E-MMT	1266	148	114	0.19
BK/E-HNT	1090	139	106	0.20
BK/E	895	133	69	0.21

## Data Availability

Not Applicable.

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
