# Peer review of "Effects of Nanoclay on Mechanical and Dynamic Mechanical Properties of Bamboo/Kenaf Reinforced Epoxy Hybrid Composites"

_polymers, 2021, doi:10.3390/polym13030395_

Round 1

Reviewer 1 Report

- Page 1 (Abstract), change “98.4 %, 41.5 %, and 21.7%”for “98.4, 41.5, and 21.7”

- Manuscript has some typos, revise carefully and correct it

- 2.1 Materials, include information about purifications materials or used as received

- Page 3 Change”65.7 % and 72.6 118 %. Hemicellulose content is 17.8 % and 11.1 %; lignin content is 6.0 % and 9.5 %” for “65.7  and 72.6 118 %. Hemicellulose content is 17.8 and 11.1 %; lignin content is 6.0 and 9.5 %”

- Line 145, include information if figure 1 is SEM or what

- Line 189, changer “19.12 %, 13.58 %, and 44.5 %”for “19.12, 13.58, and 44.5 %”. The same recommendation for line 191

- Figure 4, change “0.00, 20.00, 40.00, 60.00, 80.00, 100.00, 120.00”, for “0, 20, 40, 60, 80, 100, 120”. The same recommendation for Figure 5 (remove two 00 after point)

- Line 296 change “21.8 %, 19.6 % and 54.7%”for “21.8, 19.6 and 54.7%”. Line 298 change “27.0 % and 24.4 %” for “27.0 and 24.4 %”

- Figure 6, remove two 00 after point “0.00, etc.”

- Line 355 and 356 change “98.4 %, 355 41.5 %, and 21.7%”for “98.4, 355 41.5, and 21.7%”

- Line 394 change “159 %, 65.2 %” for ”159, 65.2”

- Manuscript has some interesting results but need to improve discussion for all Figures

- All Figures have poor resolution, improve it at minimal 300 dpi

Author Response

Thanks for your comments. We revise manuscript as per your comments. attached reply to reviewer comments

Reviewer 2 Report

REVIEW

on article

Effects of nanoclay on mechanical and dynamic mechanical properties of Bamboo/Kenaf Reinforced Epoxy Hybrid Composites

Siew Sand Chee, Mohammad Jawaid, O.Y. Alothman and H. Fouad

SUMMARY.

Researchers have recently shown strong interest in nanocomposites based on clay or epoxy resins. This interest is due to the positive influence of these products on the mechanical, thermal, protective and anti-corrosion properties of coatings. Since products based on epoxy resins are characterized by high physical, mechanical, dielectric and chemical properties and are widely used in various industries, they serve as an excellent matrix for obtaining nanocomposites.

The article is devoted to the topical issue of finding a new mixture of composite materials based on epoxy resins. This work studies the mechanical and dynamic mechanical properties of non-woven hybrid composites made of bamboo / woven kenaf / epoxy resin filled with nanoclay, which were prepared by dispersing. The authors investigated the effect of adding nanoclay on the tensile, bending, and impact properties of hybrid nanocomposite samples. The fractography of the tensile fractured samples was investigated using a field emission scanning electron microscope.

The research is experimental. The authors made samples with an amount of nanoclay of 1 wt%. The authors showed that the mechanical properties of hybrid composites in tension, bending and impact are improved by the inclusion of nanoclay. Hybrid composites exhibit better mechanical properties with 44.5% improvement in tensile strength, 61% in flexural strength and 54.7% in toughness over baseline.

Therefore, an article devoted to the study of new composite materials is relevant and of scientific interest.

Specific comments.

  1. In the final part of the Introduction, I recommend that the authors clearly formulate the purpose of the study and remove the results presented in the Conclusion.
  2. Line 178-180. This is a well-known fact. It is better to remove the phrase.
  3. How many samples have you tested?
  4. Line 183-184. "With the addition of nanoclay, increases the stress but reduces the strain". Need to rephrase. In itself, the addition of nanoclay does not increase stress. Nanoclay increases the material's ability to withstand greater stress, but overall deformation is reduced. That is, the material becomes fragile.
  5. Section 3.5 What frequencies were the dynamic mechanical analysis performed at?
  6. Why are the frequency-dependent characteristics of the tested samples not shown?
  7. Section 3.5.2 How can you explain the fact that the Loss Modulus rises sharply in the temperature range from 60ºC to 80°C? And in other temperature regions, it is constant.

The authors have done a great job studying the properties of the proposed materials in tension-compression, bending, impact strength. Large-scale work in dynamic mechanical tests at various temperatures and analysis of the results with comparison with the data obtained by other authors. This will undoubtedly attract the interest of readers.

In general, I recommend the article for publication after minor corrections.

Author Response

Thanks for your valuable comments and revised manuscript as per your suggestion. 

Here attached Reply to Reviewer comment
